# HDAC1,2 Knock-Out and HDACi Induced Cell Apoptosis in Imatinib-Resistant K562 Cells

**DOI:** 10.3390/ijms20092271

**Published:** 2019-05-08

**Authors:** Shu-Huey Chen, Jyh-Ming Chow, Yao-Yu Hsieh, Chun-Yu Lin, Kai-Wen Hsu, Wen-Shyang Hsieh, Wei-Ming Chi, Beished M. Shabangu, Chia-Hwa Lee

**Affiliations:** 1Department of Pediatrics, School of Medicine, College of Medicine, Taipei Medical University, Taipei 11031, Taiwan; Shu117@tmu.edu.tw; 2Department of Pediatrics, Shuang Ho Hospital, Taipei Medical University, New Taipei City 23561, Taiwan; 3Department of Hemato-Oncology, Wan Fang Hospital, Taipei Medical University, Taipei 11696, Taiwan; b681036@tmu.edu.tw; 4Division of Hematology and Oncology, Shuang Ho Hospital, Taipei Meidcal University, New Taipei City 23561, Taiwan; 10573@s.tmu.edu.tw; 5Institute of Bioinformatics and Systems Biology, National Chiao Tung University, Hsinchu 30068, Taiwan; chunyulin.bi99g@g2.nctu.edu.tw; 6Bioinformatics Center, Institute for Chemical Research, Kyoto University, Kyoto 611-0011, Japan; 7Graduate Institute of New Drug Development and Biomedical Sciences, China Medical University, Taichung 40402, Taiwan; kwhsu@mail.cmu.edu.tw; 8Research Center for Tumor Medical Science, China Medical University, Taichung 40402, Taiwan; 9Department of Medical Laboratory, Shuang Ho Hospital, Taipei Medical University, Taipei 23561, Taiwan; 12638@s.tmu.edu.tw; 10Department of Clinical Pathology, Shuang Ho Hospital, Taipei Medical University, Taipei 23561, Taiwan; rc4202@tmu.edu.tw; 11School of Medical Laboratory Science and Biotechnology, College of Medical Science and Technology, Taipei Medical University, Taipei 11031, Taiwan; shabangu.beished@gmail.com; 12Ph.D. Program in Medicine Biotechnology, College of Medical Science and Technology, Taipei Medical University, Taipei 11031, Taiwan; 13TMU Research Center of Cancer Translational Medicine, Taipei 11031, Taiwan

**Keywords:** imatinib, CML, histone deacetylase inhibitor, imatinib-resistant, CRISPR/Cas9

## Abstract

Since imatinib (Glivec or Gleevec) has been used to target the BCR-ABL fusion protein, chronic myeloid leukemia (CML) has become a manageable chronic disease with long-term survival. However, 15%–20% of CML patients ultimately develop resistance to imatinib and then progress to an accelerated phase and eventually to a blast crisis, limiting treatment options and resulting in a poor survival rate. Thus, we investigated whether histone deacetylase inhibitors (HDACis) could be used as a potential anticancer therapy for imatinib-resistant CML (IR-CML) patients. By applying a noninvasive apoptosis detection sensor (NIADS), we found that panobinostat significantly enhanced cell apoptosis in K562 cells. A further investigation showed that panobinostat induced apoptosis in both K562 and imatinib-resistant K562 (IR-K562) cells mainly via H3 and H4 histone acetylation, whereas panobinostat targeted cancer stem cells (CSCs) in IR-K562 cells. Using CRISPR/Cas9 genomic editing, we found that HDAC1 and HDAC2 knockout cells significantly induced cell apoptosis, indicating that the regulation of HDAC1 and HDAC2 is extremely important in maintaining K562 cell survival. All information in this study indicates that regulating HDAC activity provides therapeutic benefits against CML and IR-CML in the clinic.

## 1. Introduction

Chronic myeloid leukemia (CML) is a hematological disease with reciprocal translocation between the break-point cluster (*BCR*) gene on chromosome 22 and the Abelson leukemia virus oncogene (*ABL*) gene on chromosome 9, also termed the Philadelphia (Ph) chromosome. The BCR-ABL fusion protein constitutively activates tyrosine kinase activity and induces many downstream signaling pathways, such as the AKT and mitogen-activated protein kinase (MAPK) pathways, contributing to cell proliferation and resistance to cell apoptosis and causing the disruption of genetic stability [1]. With the discovery of effective tyrosine kinase inhibitors (TKIs), such as imatinib (Glivec or Gleevec) and its derivatives dasatinib, nilotinib, bosutinib, and ponatinib, CML has become a manageable chronic disease with a long-term survival exceeding 85% [2]. The inhibitory mechanism of imatinib mesylate involves binding to the ATP-binding pocket of BCR-ABL, preventing substrate phosphorylation and activation from BCR-ABL. However, in the clinic, imatinib resistance has become a major problem for CML treatment, mainly as a consequence of BCR-ABL mutations, BCR-ABL overexpression and other BCR-ABL-independent pathways. Accordingly, approximately 15%–20% of these CML patients ultimately develop resistance to imatinib and then progress to an accelerated phase and eventually to a blast crisis [3]. Thus, overcoming imatinib resistance has attracted increased attention during the treatment of CML.

The histone deacetylase (HDAC) family is a group of proteins that maintains the acetylation/deacetylation balance of histones and nonhistone proteins, resulting in epigenetic regulation of gene expression by changing the structure of chromatin and modulating the accessibility of transcription factors to their target DNA sequences [4]. Deacetylation of histones causes chromatin condensation, while decondensation is caused by increased acetylation [5], implying that this DNA remodeling might result in decreased or increased gene transcription for cell survival or cell death. Some studies have noted that histone deacetylase inhibitors (HDACis) transcriptionally activate CDKN1A (known as p21) and force cells to undergo cell cycle arrest and apoptosis [6]. In the clinic, five FDA-approved HDACis have been used against peripheral T-cell lymphoma, multiple myeloma or even bipolar disorders [7]. For other chemotherapies, HDACis provide optimal benefits in combinatory schedules, showing high therapeutic efficacy on cancer therapy, such as triple negative breast cancer [8,9] and acute myeloid leukemia [10].

Epigenetic therapy has been proven to be a successful approach for the treatment of several human malignancies, including liver [11], blood [12,13], lung [14] and colon [15] cancers. In the present study, we investigated whether FDA-approved HDACi drugs (panobinostat, belinostat, vorinostat, and valproic acid) show anticancer activity on both CML (K562) and imatinib-resistant CML (IR-K562) cells. These experiments helped us to reveal the potent mechanism that induced drug resistance in CML. Furthermore, we developed a powerful tool combining lentivirus transfection and a noninvasive apoptosis detection sensor (NIADS), which has the advantages of being easy to handle and allowing quantitative and kinetic analyses of apoptotic cell death [8]. Here, we will not only use this platform to detect HDAC-induced apoptosis through in vivo image system (IVIS) observation but also further apply NIADS detection to flow cytometry measurements for individual cell analysis. This strategy will largely improve the detection of this live-cell-based apoptosis detection platform. Furthermore, using genomic editing of clustered regularly interspaced short palindromic repeat (CRISPR/Cas9) targeting HDAC genes, we tried to elucidate the regulatory mechanism behind epigenetic gene alternation and prevent side effects in future HDACi drug design. In this study, using HDAC inhibitors that induce selective chromatin remodeling events to alter specific gene expression patterns with accurate dosage settings and specificity will trigger a new wave of drugs with refined strategies for the treatment of human CML or IR CML malignancies.

## 2. Results

### 2.1. FDA-Approved HDACi-Induced K562 Cell Apoptosis

CML (K562) cells were treated with different concentrations of four FDA-approved clinical HDACi drugs (panobinostat, belinostat, vorinostat, and valproic acid) for the cell viability assay (Figure 1A,B). After 24 h of drug exposure, panobinostat had the lowest cell viability at 0.1 μM (27.6%) treatment compared to the belinostat (68.5%), vorinostat (95%), and valproic acid (98.1%) treatments. In addition, with increasing panobinostat concentrations, K562 cells showed the same cell viability at 1 and 10 μM (18.8% and 16.5%), whereas belinostat and vorinostat reached the maximum anticancer effect at 10 μM (15.7% and 21.6%) treatments, respectively. Using a viability assay (Figure 1C), we found that the IC50 of panobinostat was 0.04 μM, whereas belinostat and vorinostat had IC50 concentrations of 1.4 and 2.94 μM, respectively.

In our previous study, we established a bioluminescence-based live cell NIADS system to evaluate the quantitative and kinetic analyses of apoptotic cell death [8,16]. Using this assay, we determined apoptotic events by simply measuring bioluminescence activities from live cells. Here, we used the NIADS system to assess stably expressing K562 cells (NIADS-K562) treated with various concentrations of HDACis for 24 h and measured bioluminescence activity by IVIS (Figure 2A). The bioluminescence activities from NIADS-K562 showed a significant increase after the 0.1 μM panobinostat, belinostat and vorinostat treatments compared with dimethyl sulfoxide (DMSO) treatment. Next, we want to know whether or not the apoptotic cell population in NIADS platform can be quantified by flow cytometry. To achieve that, we measured the spectrum of bioluminescence of NIADS through the light wavelength from 400 to 750 nm (Figure 2B). The wavelength spectrum clearly demonstrated that the bioluminescence emitted photons at 560 to 620 nm, which is applicable for the FL2 channel of flow cytometry (564–606 nm, red column), whereas the FL1 channel detects green fluorescence (515–545 nm, green column). Next, using FL2 channel flow cytometry for bioluminescent detection, we found that panobinostat and belinostat induced significant cell apoptosis of 90.4% and 58.6% in total NIADS-K562 cells, whereas DMSO, vorinostat and valproic did not induce apoptosis (Figure 2C).

### 2.2. HDACi Induced Histone Acetylation and Apoptosis-Related Protein Expression

We next examined the acetylation sites of the histone complex that would be activated by HDACi drugs on K562 cells (Figure 3A). After 6 h of HDACi treatment, panobinostat significantly induced histone H3 acetylation at amino acids 9, 18 and 56, whereas histone H4 was acetylated at sites 8 and 16; vorinostat and belinostat had mild acetylation effects at the H3 and H4 acetylation sites. In the deeper apoptosis investigation, we found that panobinostat induced dramatic p21 induction and activated apoptosis signaling cascades, including Caspase 3 and PARP activation, whereas vorinostat and belinostat treatments of K562 cells showed weak p21, Caspase 3 and PARP activation. This evidence indicated that HDACi-induced H3 and H4 acetylation may not be the only factor that triggers cell cycle arrest and apoptosis in CML cells. Other panobinostat-induced functional changes may be involved in the anticancer activity. Next, we used a fluorescence-based live/dead cell viability assay to investigate the dose-dependent nature of panobinostat-induced apoptosis in K562 cells. After 24 h of panobinostat treatment, PI-stained dead cells (red) were significantly induced in the 0.1, 1 and 1 μM treatment groups (*p* < 0.05 at 0.1 μM treatment, *p* < 0.01 at 1 and 1 μM treatment), whereas the calcein AM-stained live cells (green) were gradually reduced compared to DMSO-treated K562 cells.

### 2.3. Protein Analysis of Panobinostat-Treated K562 and K562-IR Cells

In clinical analyses, imatinib has provided a dramatic improvement in outcomes for CML patients, increasing the five-year survival rate from 45% to more than 80%. However, resistance or failure to respond to imatinib treatment has emerged as a significant clinical problem affecting approximately 1/3 of all CML patients and leading to cancer progression. To find an alternative treatment for imatinib-resistant CML patients, we developed a K562-IR cell line. To reveal the drug sensitivity to imatinib of either K562-IR or K562 cells, we added various concentrations of imatinib from 0.01 μM to 10 μM for 24 h (Figure 4A). Using the MTT cell viability assay, we found that K562-IR showed extremely high cell survival when exposed to imatinib at 0.1, 1 and 10 μM compared to K562 cells (*p* < 0.01). The IC50 values of imatinib on both K562-IR and K562 are 2.796 μM and 0.093 μM, respectively, confirming the imatinib-resistant character of K562-IR (Figure 4C). However, with various concentrations of panobinostat treatment, we found that both K562-IR and K562 cells had significant decreases in cell viability after 0.1 μM treatment (Figure 4B). The IC50 values of panobinostat for both K562-IR and K562 were 0.2032 μM and 0.0385 μM, implying that panobinostat therapy would also be applicable for imatinib-resistant patients in the clinic.

We next investigated whether panobinostat-induced histone acetylation and apoptosis signals in K562 cells differed from those in K562-IR cells (Figure 4D). After treatment with various concentrations of panobinostat on K562 cells for 6 h, we found that 0.01 μM panobinostat significantly induced histone H3 acetylation at the 9, 18 and 56 amino acids, whereas histone H4 was acetylated at sites 8 and 16 (Figure 4E). In contrast, panobinostat activated these acetylation sites on the H3 and H4 histone proteins at 0.1 μM in IR-K562 cells (Figure 4E). Furthermore, p21 expression, activated Caspase 3 and PARP were strongly induced in 0.01 μM panobinostat-treated K562 cells, whereas these proteins increased in 0.1 μM panobinostat-treated K562-IR cells. These results clearly showed that even though K562-IR cells may have low sensitivity to panobinostat compared to K562 cells, panobinostat treatment may still be used in imatinib-resistant CML cells, which had with a low IC50 of 0.2032 μM.

### 2.4. HDAC1 and HDAC2 Gene Knockout through the CRISPR/CAS9 System

Next, we investigated the utility of CRISPR/Cas9 genome editing by targeting two custom-designed protospacers on *HDAC1* on chromosome 1 and the *HDAC2* locus on chromosome 6 with a lentivirus delivery system using the MIT CRISPR Design website (http://crispr.mit.edu) with the sequence of *HDAC1* (NM_004964.2) and *HDAC2* (NM_001527.3). As shown in the *HDAC1* genomic map (Figure 5A), the protospacer 1 sgRNA targets the negative strand, and the protospacer 2 sgRNA targets the plus strand of the exon 2 *HDAC1* gene. Transduction of K562 cells with the scrambled target (SC) lentivirus produced a wild-type *HDAC1* sequence, as assessed by Sanger sequencing (Appendix A), with no evidence of gene editing. However, K562 cells transduced with *HDAC1*-1 gene-edited lentivirus (Appendix A) had multiple gene disruptions at the predicted cleavage sites (red arrowhead) compared to K562 cells transduced with the *HDAC1*-2 gene-edited lentivirus (Appendix A). In addition, through TIDE analysis, lentivirus infection with *HDAC1*-1 gene-edited cells (Figure 5B) had stronger gene editing efficiency than *HDAC1-2* gene-edited cells (Figure 5C), with 98.5% and 14.2% of the cell pool edited, respectively. The most frequent mutation in the *HDAC1*-1 gene-edited cell pool was other mutations (85.2%), whereas the frequently predicted mutation in the *HDAC1*-2 gene-edited cell pool was 1-bp insertions (4.4%). Compared to *HDAC1*-2 gene-edited cells, K562 cells transduced with the *HDAC1*-1 gene showed more significant gene disruptions in the targeted regions, with mutations primarily at the predicted cleavage sites (Appendix A). However, both protospacer 1 sgRNA and protospacer 2 sgRNAs target the plus strand of the exon 1 of the *HDAC2* gene. Sanger sequencing showed no evidence of gene editing in SC lentivirus-transduced K562 cells (Appendix A). Compared to *HDAC2*-2 gene-edited lentivirus (Appendix A)-transfected K562 cells, *HDAC2*-1 gene-edited cell lentivirus cells (Appendix A) showed significant multiple gene disruptions at the predicted cleavage sites (red arrowhead). With TIDE analysis, lentivirus infection of the *HDAC2*-1 gene-edited virus (Figure 5G) also showed stronger gene editing efficiency than the *HDAC2*-2 gene-edited population (Figure 5H), with 84% and 2.4% of the cell pool edited, respectively. The most frequent mutation in the *HDAC2*-1 gene-edited cell pool (Figure 5I) was other mutations (69%), whereas the most frequently predicted mutation in the *HDAC2*-2 gene-edited cell pool (Figure 5J) was a 1-bp insertion (1.2%). In addition, only *HDAC2*-1 gene-edited cells caused significant gene disruptions in the targeted regions, whereas no gene disruptions were found in *HDAC2*-2 gene-edited K562 cells, with mutations primarily at the predicted cleavage sites (Appendix A).

### 2.5. HDAC1 and HDAC2 gene knockout Promotes K562 Cell Apoptosis

Next, we evaluated HDAC1 and HDAC2 protein expression by western blotting of the gene-edited K562 cells (Figure 6A,B). As shown in the above genomic results, the protein levels of both *HDAC1* and *HDAC2* sgRNA-introduced K562 cells were significantly decreased compared to those of SC virus-transfected cells. In addition, *HDAC1* gene-edited cells showed increased HDAC2 protein expression, whereas *HDAC2* gene-edited cells showed increased HDAC1 protein expression in K562 cells. This observation proves that HDAC1 and HDAC2 have complementary effects. Next, we wanted to determine the biofunctional effect in *HDAC1* and *HDAC2* gene-edited K562 cells. The apoptosis measurement (determined by sub-G1) by flow cytometry showed a significant elevation in *HDAC2*-1 gene-edited cells, with 16.4%, followed by *HDAC1*-1 gene-edited cells, with 9.6% of all cell populations, whereas SC cells and *HDAC1*-2 and *HDAC2*-2 gene-edited cells remained unchanged (Figure 6C). In the signal transduction and apoptosis molecule analysis (Figure 6D), we found that the phospho-AKT and ERK proteins were inhibited in both *HDAC1*-1 and *HDAC2*-1 gene-edited K562 cells, whereas p21, cleavage PARP and Caspase 3 were significantly induced. Furthermore, we discovered that both AKT phosphorylation and the activated forms of PARP and Caspase 3 showed stronger effects in *HDAC2*-1 than *HDAC1*-1 gene-edited cells. Through flow cytometry and protein analysis, we found that the HDAC2 protein may play an important role in preventing cell apoptosis during CML disease development.

### 2.6. Panobinostat Targets CSCs of K562-IR Cells

In the above experiment, pan-HDACi panobinostat activated several H3 and H4 acetylation sites, such as H3K9 H3K18, H3K56, H4K8 and H4K16, causing dramatic apoptosis of both K562 and K562-IR cells. However, more data were needed to determine whether panobinostat achieves similar anticancer effects in both K562 and K562-IR cells (Figure 6E). Thus, we found that K562 IR cells showed more phospho-AKT and ERK protein expression and less cleaved PARP and Caspase-3 proteins than K562 cells (lane 1 vs. lane 3). After 0.1 μM panobinostat treatment for 24 h, K562 showed extreme decreases in phospho-AKT and ERK, whereas p21, cleaved PARP and Caspase 3 were significantly increased (lane 1 vs. lane 2). In contrast, in the same panobinostat treatment, K562-IR cells showed less signal transduction inhibition and apoptosis induction than panobinostat-treated K562 cells (lane 2 vs. lane 4). Based on this observation, we suspected that the drug resistance from K562-IR may be due to an increase in the cancer stem cell (CSC) population. Using real-time PCR (Q-PCR), we found that CML CSC genes, such as *ALDH1*, were dramatically elevated in K562-IR cells compared with K562 cells (Figure 6F). Following treatment of 0.1 μM panobinostat for 24 h, the *ALDH1* gene expression was significantly decreased in K562-IR cells and was close to the gene expression of K562 cells. These data indicated that long-term imatinib treatment induces CSC accumulation in K562-IR cells, which causes decreased drug sensitivity to both imatinib and panobinostat treatments. However, panobinostat treatment seems to target CSCs and reduce the CSC population in K562-IR cells, suggesting that panobinostat can be applied to prevent imatinib resistance.

## 3. Discussion

Histone acetylation has been shown to be an important regulatory mechanism that controls transcription of approximately 2%–10% of genes [4]. In fact, histones are not the only proteins that can be activated by acetylation; many proteins are positively or negatively regulated by epigenetic regulation, such as chromatin remodeling proteins, DNA-binding nuclear receptors, DNA repair enzymes, signaling mediators, structural proteins, transcription coregulators, and DNA-binding transcription factors [17,18]. Microarray analyses showed that HDACis manipulate genes in several biofunctional classes, such as cell cycle inhibition [19], β-catenin-related signaling, lineage-specific differentiation [20] and apoptotic-related cell death. Previous studies have shown that HDACis transcriptionally activated *p21* promoter-associated histones by either acetylation or methylation, whereas these factors were not altered in a multiple myeloma cell line [21]. As shown in this study, p21 was significantly induced in panobinostat-treated K562 cells, further promoting apoptosis-related proteins, such as PARP and caspases. In addition, gene edited of either the *HDAC1* or *HDAC2* gene showed that HDAC2 may play a more important role in regulating the cell cycle and apoptosis than HDAC1, with elevation of p21, c-PARP and c-Caspase 3 protein expression. HDAC2 may also be involved in cell proliferation inhibition since p-AKT was dramatically suppressed in *HDAC2* gene edited cells, whereas p-AKT expression in *HDAC1* gene edited cells remained unchanged.

HDAC1 and HDAC2 are closely related mammalian histone deacetylases that appear to mediate complementary functions in transcriptional regulation at specific sites in chromatin [22]. This complementary effect between HDAC1 and HDAC2 was also confirmed in this study. To investigate the functions of HDAC1 and HDAC2 in CML, we used CRISPR/Cas9 to specifically gene edit *HDAC1* and *HDAC2* through lentivirus transfection of K562 cells. The results clearly showed that the HDAC proteins were correspondently suppressed in both *HDAC1* and *HDAC2* sgRNA introduced cells. This highly effective gene editing result was very similar to what we observed in breast and thyroid cancer cells [23,24]. Interestingly, in *HDAC1* and *HDAC2* sgRNA introduced cells, we observed a strong protein complementary effect in both *HDAC1* and *HDAC2* gene edit K562 cells. This finding implies that HDAC absent CML may try to retain HDAC activity by increasing the protein level of the other HDAC member. However, even with this complementary effect, *HDAC1* and *HDAC2* gene edit cells still showed dramatic apoptotic cell death, indicating the important roles of HDAC1 and HDAC2 in maintaining cell survival. In the future, anticancer drugs targeting HDAC may be used as a strategy to design specific HDAC inhibitors, preventing the side effects from the current pan-HDACi treatment.

In a previous study [25], a panobinostat and ponatinib combination synergistically inhibited imatinib-resistant CML through BCR-ABL and AKT signaling. To further investigate the mechanism behind this finding, we used HDAC1 and HDAC2 as a model to demonstrate the anticancer activity by HDACi drugs both with or without imatinib-resistant CML cells. Surprisingly, we found that K562 cells were extremely sensitive to panobinostat treatment, with an IC50 of 0.04 μM (Figure 1C), whereas a previous study showed that K562 cells did not respond to panobinostat exposure, with an IC50 of 50 μM [25]. Despite this controversial issue, imatinib-resistant CML cells demonstrated equal or lower cell viability than K562 cells with panobinostat treatment in both studies. In addition, this study showed that HDACi treatment targets a CSC-like population in imatinib-resistant K562 cells, indicating that panobinostat can be used for CML patients who show no response to imatinib.

To the best of our knowledge, this study is the first to detect bioluminescence in flow cytometry. In a previous study, we developed a rapid and quantitative apoptosis detection assay based on caspase cleavage with a specific peptide sequence, DEVD, on living cells. We detected the bioluminescence of NIADS activation using a bioluminometer or IVIS instruments with total cell apoptosis status. However, with the help of flow cytometry, we can now identify apoptotic signals from individual cells, which is a major improvement in evaluating the apoptotic cell percentage of cells from clinical CML patients. In future studies, the next step will be to use NIADS to determine whether this platform can be applied for precision medicine in CML patients.

Although imatinib and its chemical derivatives are the best strategy for cancer therapy in CML patients, the emerging understanding that key molecules can mediate the drug resistance mechanism in disease progression provides the basis for novel therapeutic markers for cancer treatment in the clinic. The main contribution of this study is the finding that panobinostat significantly suppressed HDAC activity and forced histone acetylation, followed by cell cycle arrest and eventually cell apoptosis in both K562 and K562IR cells. Although K562IR showed a slight resistance to panobinostat treatment, the effective cell death drug dosage from panobinostat remained low. In fact, the current evidence showed that panobinostat seems to target the CSC population in K562IR cells. The combination of imatinib and panobinostat may target CML cells, as well as the CSC population, preventing cancer recurrence or drug resistance. *HDAC1* and *HDAC2* gene knockout through CRISPR/Cas9 provides evidence that both HDAC1 and HDAC2 are unique in maintaining K562 cell survival. This result contributed to our understanding of protein acetylation regulation from the HDAC family, indicating that HDAC member-specific inhibitors would cause significant CML cancer cell death, as much as the apoptotic effect was due to treatment with pan-HDAC inhibitors.

## 4. Materials and Methods

### 4.1. Cell Culture

The human leukemia K562 cell line (CML) was kindly provided by Dr. Kai-Wen Hsu, Graduate Institute of New Drug Development and Biomedical Sciences, China Medical University, Taichung, Taiwan. Imatinib-resistant K562 (IR-K562) cells were originally derived from K562 cells by treatment with 0.05 μM imatinib for 2 months and treatment with 0.1, 0.5, 1 and 5 μM imatinib for one month. While training IR-K562 cells, we changed the culture medium every week. The cells were maintained in Dulbecco’s modified Eagle’s medium: Nutrient Mixture F-12 (DMEM/F-12) (Gibco, Carlsbad, CA, USA). The cells were incubated with 10% (*v*/*v*) fetal bovine serum (FBS, Biological Industries, Israel). Supplements of 100 units/mL penicillin and 100 mg/mL streptomycin were used and cultured in a 37 °C incubator with 5.0% CO_2_.

### 4.2. MTT Cell Viability Assay

The cell viability was determined using 3-(4,5-dimethylthiazol-2-yl)-2,5-diphenyltetrazolium (MTT), which is based on the reduction of the yellow MTT to purple formazan by living cells [26,27]. In 96-well plates, 1 × 10^5^ K562 cells were seeded overnight before exposure to different concentrations of HDACis or imatinib according to the experimental protocol. After 24 h of treatment, the medium was changed to fresh medium containing 1 μg/mL of MTT. Two hours later, 100 μL of DMSO was added to each well, and the absorbance at 570 and 630 nm was determined. The cell viability percentage was calculated using a formula [Percentage viability = (Average OD of sample/Average OD of control) × 100].

### 4.3. Live/Dead Cell Viability Assay

K562 cells were seeded in 12-well plates overnight and incubated with HDACis for 24 h in normal culture conditions. The medium was removed and treated for 30 min in the dark with 1 μM calcein-AM and 10 μM propidium iodide (PI) prepared in a normal culture medium. The live fluorescence images were captured under a light wavelength of 488 nm (green emission) to show viable cells. The same image of the cells was also excited with a light wavelength of 532 nm (red emission) to show the dead cells.

### 4.4. Cellular Bioluminescence (IVIS) Assay

Bioluminescence imaging was performed with a highly sensitive, cooled CCD camera mounted in a light-tight specimen box (In Vivo Imaging System—IVIS; Xenogen, Alameda, CA, USA). K562 cells were plated and treated with HDACis at 0.1 μM and 1 μM for 12 h. The multiple-well plate was exposed to D-luciferin (1.5 μg/mL) and placed on a warmed stage inside the camera box during imaging. The light emitted from the cells was detected by the IVIS camera system, integrated, digitized, and displayed. Regions of interest on the displayed images were identified, and the total photon count was quantified using Living Image^®^ software 4.0 (Caliper, Alameda, CA, USA).

### 4.5. Flow Cytometry Analysis

K562 and NIADS K562 cells with stable expression (NIADS-K562) (5 × 10^6^ cells/dish) were plated in 6-cm dishes and exposed to different concentrations of HDACis or imatinib for 24 h. For sub-G1 apoptosis analysis, K562 cells were collected, washed once with PBS, fixed with 75% alcohol and analyzed with a sub-G1 cell population by flow cytometry (FACSCalibur, BD Biosciences, San Jose, CA, USA). For bioluminescence detection, NIADS-K562 cells were collected, washed once with PBS and exposed to D-luciferin (1.5 μg/mL) before flow cytometry analysis using FL2 channels.

### 4.6. Real-Time Quantitative Polymerase Chain Reaction (Q-PCR)

Primers for the ALDH1 region (forward 5′-GAGTGTTGAGCGGGCTAA-3′ and reverse 5′-CTCCTCCACATTCCAGTTTG-3′) were used for gene quantification. All oligo primers were synthesized by Genomics BioSci and Tech (Taipei, Taiwan). A LightCycler thermocycler (Roche Molecular Biochemicals, Mannheim, Germany) was used for Q-PCR analysis. One microliter of sample and master mix were first denatured for 10 min at 95 °C and then subjected to 40 cycles (denaturation at 95 °C for 5 s; annealing at 60 °C for 5 s; and elongation at 72 °C for 10 s) with detection of fluorescence intensity. All PCR samples underwent melting curve analysis to detect nonspecific PCR products. Gene expression from the Q-PCR analysis was normalized to *GUS* expression using the built-in Roche LightCycler Software, version 4 [24,27].

### 4.7. Protein Extraction, Western Blotting, and Antibodies

For western blot analysis, K562 and in vitro imatinib-resistant K562 (K562-IR) cells were collected and washed once with ice-cold PBS, followed by radioimmunoprecipitation assay (RIPA) lysis buffer addition, which contained protease inhibitors. Fifty microgram of protein from each sample was resolved by sodium dodecyl sulfate polyacrylamide gel electrophoresis (SDS-PAGE) and transferred to a nitrocellulose membrane. The information of primary antibodies and the secondary antibodies used in this study is provided in Appendix A. All primary antibodies were used at a 1:1000 dilution with overnight hybridization, followed by a one-hour incubation with a 1:4000 dilution of the secondary antibodies.

### 4.8. Lentivirus Production

HDAC1 and HDAC2 targeting lentiviral particles and NIADS lentivirus were produced by transient transfection of Phoenix-ECO cells (CRL-3214) using TransIT^®^-LT1 Reagent (Mirus Bio LLC, Madison, WI, USA). Guide oligonucleotides were phosphorylated, annealed, and cloned into the BsmBI site of the lentiCRISPR v2 vector (Addgene, 52961, kindly provided by Feng Zhang) according to the Zhang laboratory protocol [28] (F. Zhang lab, MIT, Cambridge, MA, USA). All plasmid constructs were verified by sequencing. The HDAC1, HDAC2 and NIADS plasmids were cotransfected with pMD2.G (Addgene plasmid #12259) and psPAX2 (Addgene plasmid #12260, both kindly provided by Didier Trono, EPFL, Lausanne, Switzerland). Lentiviral particles were collected at 36 and 72 h and then concentrated with a Lenti-X Concentrator^®^ (Clontech, Mountain View, CA, USA). The lentivirus concentration for each gene was quantified by Q-PCR.

### 4.9. Statistical Methods

All data are expressed as the mean ± SD, and Student’s *t*-test analysis was performed for the pairwise samples. All statistical comparisons were performed using SigmaPlot graphing software (San Jose, CA, USA) and Statistical Package for the Social Sciences v.13 (SPSS, Chicago, IL, USA). A *p*-value < 0.05 was considered statistically significant, and all statistical tests were two-sided.

## Figures and Tables

**Figure 1 ijms-20-02271-f001:**
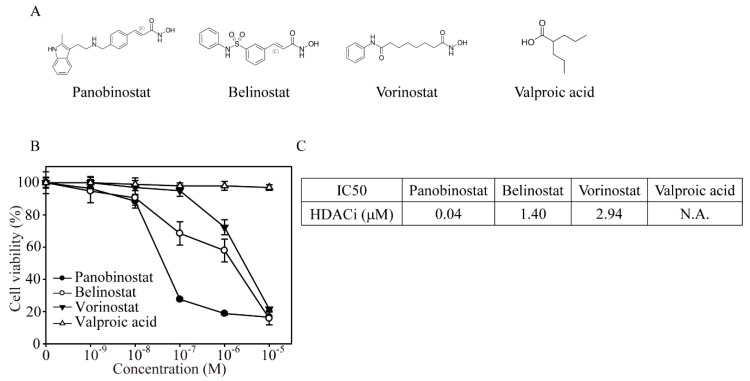
Food and Drug Administration (FDA)-approved HDACis significantly suppressed K562 cell viability in a dose-dependent manner. (**A**) Chemical structures of panobinostat, belinostat, vorinostat and valproic acid. (**B**) MTT cell viability after four HDACi (0.001, 0.01, 0.1, 1 and 10 µM) treatments on K562 cells for 24 h. The values are presented as the means and standard errors. The experiment was performed at least in triplicate. (**C**) The IC50 of HDACi is the drug concentration that induced a 50% inhibition of cell viability.

**Figure 2 ijms-20-02271-f002:**
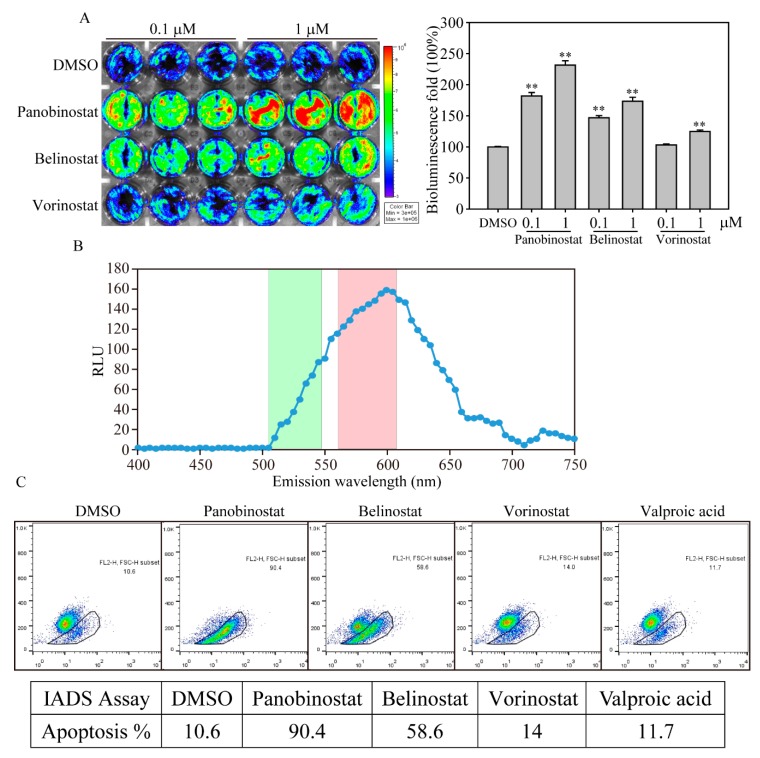
HDACi induced K562 cell apoptosis in bioluminescence-based live cell measurement. (**A**) The IVIS image of HDACi-induced cell apoptosis in NIADS stably expressed K562 (NIADS-K562) cells. The cells received HDACis or DMSO at various concentrations (0.1 and 1 µM) for 12 h and measured luciferase activity (present by photos influx). The cells in a 24-well plate were exposed to luciferin at a concentration of 1.5 μg/mL and calculated in an IVIS 200 Spectrum image system. The yellow and red colors indicate high photons of bioluminescence activity, whereas the green and blue colors indicate low photons of bioluminescence activity. The intense luciferase activity indicates apoptotic signals from NIADS. The bar figure illustrates the 100% mean and standard deviation of the photons compares to DMSO addition. Data were analyzed with Student’s *t*-test; all *p*-values were two-sided. *p*-values less than 0.01 are presented with two asterisks. (**B**) The emission wavelength spectrum of the NIADS protein. NIADS-K562 cells were exposed to luciferin, and the wavelength spectrum was measured every 5 nm from 400 nm to 750 nm. The green bar in the background indicates the fluorescent wavelength that can be detected in the FL1 channel, whereas the red bar indicates the fluorescent wavelength in the FL2 channel of flow cytometry. (**C**) Apoptosis detection in HDACi-treated NIADS-K562 cells. The cells were exposed to 1.5 μg/mL luciferin, and bioluminescence was measured in the FL2 channel of a flow cytometer. Apoptotic NIADS-K562 cells with bioluminescence were calculated in the whole cell population.

**Figure 3 ijms-20-02271-f003:**
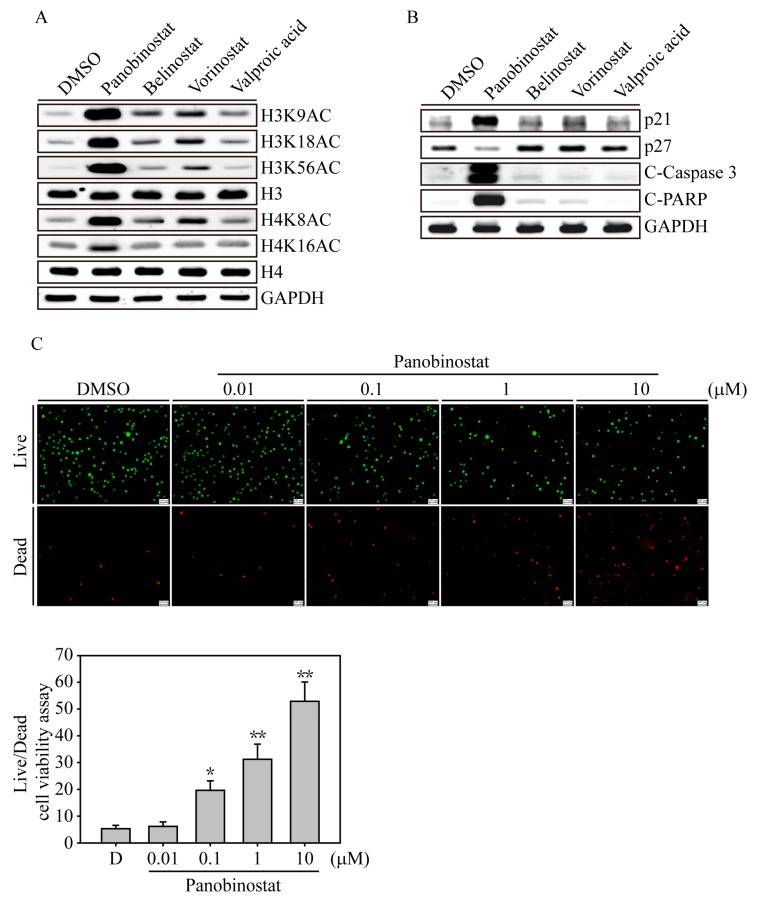
HDACi induced histone acetylation, cell cycle arrest and apoptosis-related protein expression. (**A**) K562 cells were treated with 1 µM HDACi for 6 h, and the cell lysates were immunoblotted with different H3 (H3K9AC, H3K18AC and H3K56AC) and H4 (H4K8AC and H4K16AC) histone acetylation antibodies. H3, H4 and glyceraldehyde-3-phosphate dehydrogenase (GAPDH) immunoblots served as internal controls. (**B**) K562 cell lysates treated with 1 µM HDACi for 24 h were examined for cell cycle (p21 and p27) and apoptotic-related protein (C-Caspase 3: cleaved Caspase 3 and C-PARP: cleaved PARP) expression. GAPDH immunoblotting served as an internal control. (**C**) Live/Dead cell viability assays. Fluorescence images of K562 cells exposed to different concentrations of panobinostat (0.01 to 10 µM) for 24 h. The cells were costained with 1 μM calcein-AM/10 μM PI and excited with light at 488 nm (green emission) to show viable cells. The same image of the cells also excited with 532 nm light (red emission) to show the dead cells. The scale bar on the right-bottom corner indicates 100 μM. Data are presented as the mean and standard deviation. Data were analyzed with Student’s *t*-test; all *p*-values were two-sided. *p*-values less than 0.05 are indicated with an asterisk, and values less than 0.01 are presented with two asterisks.

**Figure 4 ijms-20-02271-f004:**
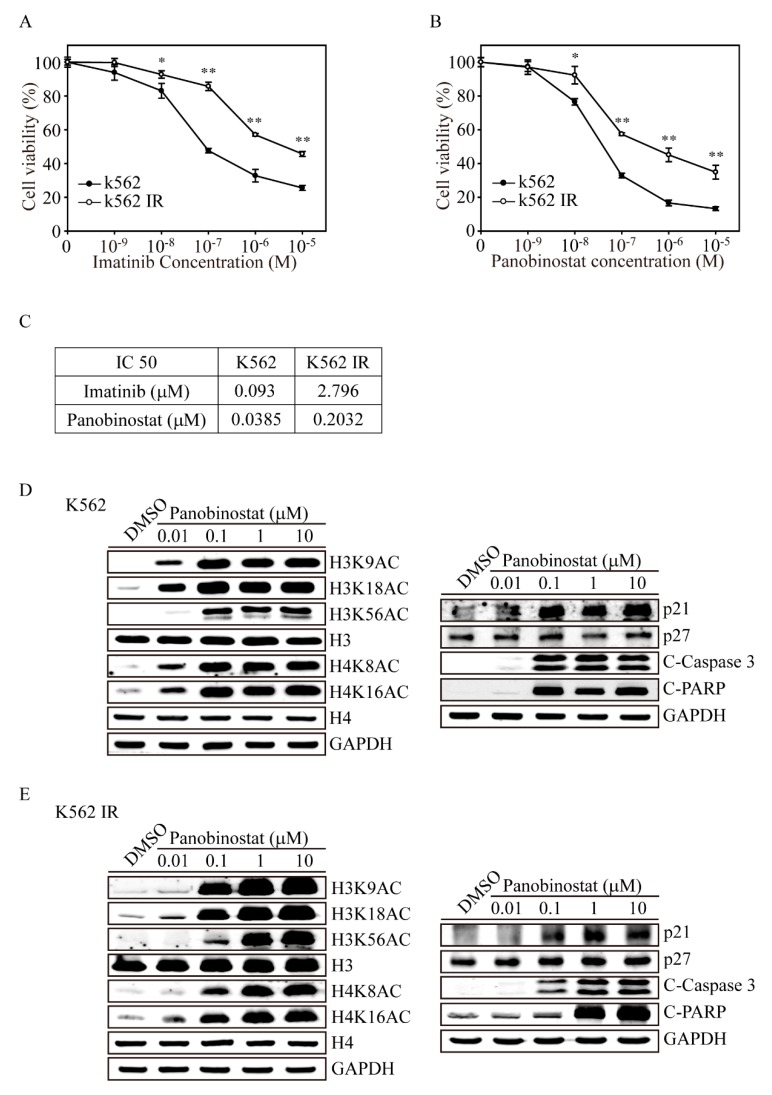
Panobinostat has anticancer effects on imatinib-resistant K562 cells. Both K562 and imatinib-resistant K562 (K562-IR) cells were seeded overnight and treated with 0.001, 0.01, 0.1, 1 and 10 μM of (**A**) imatinib or (**B**) panobinostat for 24 h. The cells were assessed for cell viability by MTT determination. Data are presented as the mean and standard deviation. Data were analyzed with Student’s *t*-test; all *p*-values were two-sided. *p*-values less than 0.05 are indicated with an asterisk, and values less than 0.01 are presented with two asterisks. (**C**) The IC50 values of imatinib and panobinostat treatments were calculated as the drug concentration that induced a 50% inhibition in cell viability. Immunoblot analyses of histone acetylation (H3K9AC, H3K18AC, H3K56AC, H4K8AC and H4K16AC), cell cycle arrest (p27 and p27) and apoptotic-related protein (C-Caspase 3 and C-PARP) expression following various panobinostat treatments for 6 h and 24 h on both (**D**) K562 and (**E**) K562-IR cells. H3, H4 and GAPDH immunoblots served as internal controls.

**Figure 5 ijms-20-02271-f005:**
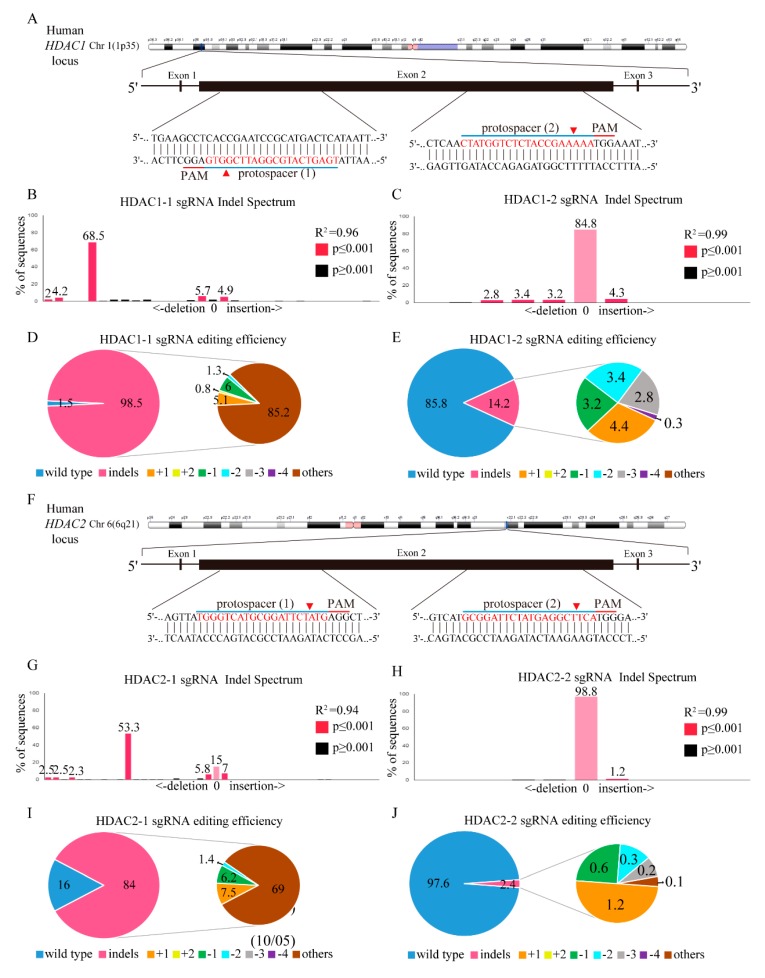
*HDAC1* and *HDAC2* gene editing in K562 cells using the CRISPR/Cas9 system. (**A**) Schematic representation of the human *HDAC1* DNA locus and two protospacer sequences (blue underline) for editing. The arrowhead indicates the expected Cas9 cleavage site. The protospacer adjacent motif (PAM, red underline) is the motif required for Cas9 nuclease activity. Scrambled (SC) and *HDAC1* gene-edited cells were delivered to K562 cells by lentivirus. After transduction, DNA from virus-infected cells was purified and subjected to Sanger sequencing of *HDAC1* exon 2. The TIDE algorithm analysis is shown for (**B**) *HDAC1*-1 and (**C**) *HDAC1*-2 gene-edited virus transfected into K562 cells compared to SC K562 cells. The pie charts show the percentages of indels in the *HDAC1* gene edited by (**D**) *HDAC1*-1 and (**E**) *HDAC1*-2 gene-edited lentivirus. The gene editing efficiency of the two-gene edited virus introduced cells is presented in pink, while the two most common other mutations and +1 are presented in brown and yellow colors, respectively. (**F**) Schematic representation of the human *HDAC2* DNA locus and two protospacer sequences (blue underline) for editing, and PAM sequences for Cas9 recognition (red underline). The arrowhead indicates the expected Cas9 cleavage site. PAM is the motif required for Cas9 nuclease activity. SC- and *HDAC2*-edited cells were delivered to K562 cells by lentivirus. After transduction, DNA from virus-infected cells was purified and subjected to Sanger sequencing of *HDAC2* exon 2. The TIDE algorithm analysis is shown for (**G**) *HDAC2*-1 and (**H**) *HDAC2*-2 gene-edited cells virus transfected into K562 cells compared to SC K562 cells. The pie charts show the percentages of indels in the *HDAC2* gene edited by (**I**) *HDAC2*-1 and (**J**) *HDAC2*-2 gene-edited lentivirus. The gene editing efficiency of the two gene-edited cells is presented in pink, while the two most common other mutations and +1 are presented in brown and yellow colors.

**Figure 6 ijms-20-02271-f006:**
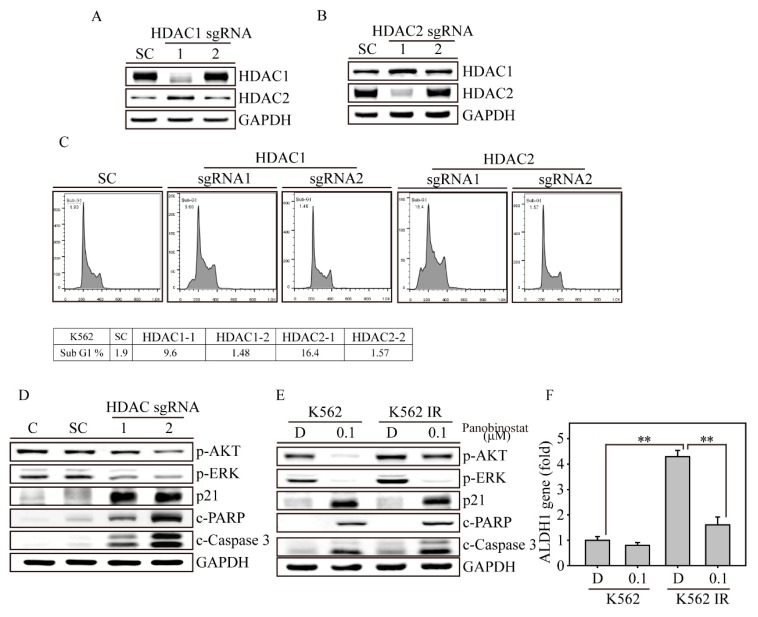
HDAC1 and HDAC2 inhibition induced cell apoptosis in both K562 and K562-IR cells. Immunoblot analysis of protein expression in (**A**) *HDAC1* and (**B**) *HDAC2* gene-edited K562 cells. (**C**) *HDAC1* and *HDAC2* gene-edited K562 cells were analyzed for a sub-G1 population by flow cytometry. (**D**) *HDAC1* and *HDAC2* gene-edited cells were analyzed for signal transduction (p-AKT and p-ERK), cell cycle (p21) and apoptosis-related protein expression compared to control and SC K562 cells. (**E**) K562 and K562IR cells with or without 0.1 μM panobinostat treatments were analyzed for signal transduction (p-AKT and p-ERK), cell cycle (p21) and apoptosis-related protein expression. GAPDH immunoblotting served as an internal control. (**F**) *ALDH1* gene expression was analyzed in K562 and K562IR cells with or without 0.1 μM panobinostat treatment for 24 h. The gene expression was measured by real-time PCR (Q-PCR). Data are presented as the mean and standard deviation and analyzed with Student’s *t*-test. All *p*-values were two-sided, and *p*-values less than 0.01 are presented with two asterisks.

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
