# Peer review of "HDAC1,2 Knock-Out and HDACi Induced Cell Apoptosis in Imatinib-Resistant K562 Cells"

_ijms, 2019, doi:10.3390/ijms20092271_

Round 1
Reviewer 1 Report
My comments are addressed.
Author Response
Thank you for the comments.
Reviewer 2 Report
The manuscript by Chen et al. entitled “HDAC1, 2 Knock-out and HDACi Induced Cell Apoptosis in Imatinib-resistant K562 Cells” describe the identification of HDAC inhibitors as agents that sensitize CML cells K562 to cell death. They show that this sensitization occurs by increased levels of p21, c-PARP, and cCaspase 3, which also occur simultaneously with increased levels of H3/H4 acetylation. These changes in cell death proteins can also be observed in cells where HDAC1 or HDAC2 have been inactivated by CRISPR/Cas9. The authors have improved the manuscript a lot and it is now more clear than in the past. I would encourage the authors to address the following minor comments before publication.
Minor comments:
1) The micro sign is missing in the description of molarity (concentration) p. 3, ln. 112
2) Figure 1B, the symbols/marks in the lines of the graph are unreadable and the reader is unable to determine which compound had which effect.
3) The discussion regarding Fig. 2A on p. 5, ln. 191-195 seems not to represent what is observed on the figure.
4) The values of the bioluminescence shown on Figure 2A have no statistical significance and are thus difficult to interpret.
5) Figure 2A has a typo “orinostat”
6) Figure 2C the flow plots are too small, Vorinostat is labeled twice
Author Response
The response to reviewer.
1. The micro sign is missing in the description of molarity (concentration) p. 3, ln. 112
Response: The correct concentration has been labeled. Please see page3, line 113 in the revised manuscript.
2. Figure 1B, the symbols/marks in the lines of the graph are unreadable and the reader is unable to determine which compound had which effect.
Response: The figure 1B has been redrawn (enlarge the symbols/marks). Please see page 5, figure 1B in the revised manuscript.
3. The discussion regarding Fig. 2A on p. 5, ln. 191-195 seems not to represent what is observed on the figure.
Response: The inappropriate sentence has been removed. Please see page 5, line 191-194 in the revised manuscript.
4. The values of the bioluminescence shown on Figure 2A have no statistical significance and are thus difficult to interpret.
Response: The statistical significance in figure 2A has been labeled (compares to DMSO control), as well as the result description in figure legend. Please see page 6, figure 2A and line 210-212 in the revised manuscript.
5. Figure 2A has a typo “orinostat”
Response: The correct “Vorinostat” has been labeled in figure 2A. Please see page 6, figure 2A in the revised manuscript.
6. Figure 2C the flow plots are too small, Vorinostat is labeled twice
Response: The flow plots has been enlarged in figure 2C Please see page 6, figure 2A in the revised manuscript.

Reviewer 3 Report
The changes were made in the revised version of the manuscript according to the recommendations and suggestions. The authors repeated the experiments and updated the figures. The manuscript in the present form is suitable for publication.
Author Response
Thank you for the comments.
This manuscript is a resubmission of an earlier submission. The following is a list of the peer review reports and author responses from that submission.
Round 1
Reviewer 1 Report
The manuscript entitled "HDAC1,2 Knock-out and HDACi Induced Cell Apoptosis in Imatinib-Resistant K562 Cells" by Shu-Huey Chen et al. describes how HDAC inhibitors work to induce apoptosis and to suppress growth of CML cells with and without resistance to imatinib. The work is relevant because CML patients develop resistance to imatinib and this significantly limits the utility of this drug. However, very similar studies have already been reported (Matsuda et al., Cancer Sci., 2016) and these results are not discussed anywhere in the paper. Matsuda et al. describe how panobinostat works in K562 cells with and without resistance to imatinib. This is the same drug Shu-Huey Chen et al. investigate in the same cell line.
Figure 3C should have the Y-axis edited to say "Dead/Live cell percentage" to match the text.
Without single cell sequencing, it is hard to conclude that panobinostat affects the small population of cancer stem cells in the K562 cell line.
The results should be discussed in the context of previously reported findings.
The manuscript requires careful proofreading.
Reviewer 2 Report
The manuscript by Chen, et al. describes the finding that HDAC inhibitors enhance the cell kill effect of Imatinib in CML cells. Importantly, this enhanced cell killing effect also occurs in cells that have developed resistance to Imatinib, which occurs in ~33% of CMLs. The findings could be potentially important and of interest. However, the manuscript is very poorly written, the English level is not appropriate for publication, the majority of the axes of the figures are impossible to read. Moreover, the authors show contradicting data in the manuscript (levels of cCaspase-3, cPARP, p21 on F.4D,E vs. F.6D,E). Finally, the authors use CRISPR/Cas9 to inactivate the HDAC1/2 genes and their western blots show that there is still protein present, which argues against their claims. Single cell clones of cells with the inactivated gene should be used in the experiments in order to make the results clear. As it is these cells cannot be called HDAC1/2 KOs. There are many more mistakes throughout the manuscript that are too numerous to list (e.g., nM instead of nm, histone H2 instead of H4, many others).
Reviewer 3 Report
The manuscript describes whether the inhibition of histone deacetylase might be considered as a perspective therapeutic benefits against imatinib-resistant CML.
I have the following concerns and suggestions regardiung the manuscript:
The manuscript requires a substantial editing of English language and style. Sometimes it's very difficult to understand what the authors are trying to say. For example: the lines 170-171 - After 24 hours drug exposure, panobinostat had the highest anti-cancer activity at 0.1 M (27.6%), compares to belinostat (68.5%), vorinostat (95%) and valproic acid (98.1%) treatments. It's unclear what are the numbers mean (IC50, % of the vital cells or something else)?
Figure 2A is a bit confusing... The authors mentioned that the intense of luciferase activity indicates apoptosis signals from NIAD.S (lines 212-213). If this point is correct (I believe that this is correct due to the DMSO-treated data shown on Fig 2), the authors should explain why the lower (0.1 mM) concentrations of HDAC-inhibitors (in particular, panobiostat and belinostat) exhibited much more potent pro-apoptotic activities in K562 cells when compared to the higher concentrations of the drugs used in present study (1 mM). The authors are trying to explain this fact by the potent anti-cancer activities of higher doses of these drugs that led to complete death of K562 cells treated with 1mM (no live cells in culture, as was mentioned in lthe text - ines 192-194). However, IC50 data shown on Figure 1C indicates that IC50 values for these compounds were between 1 - 3 mM. This makes the argument about the abscence of live cells in cultures treated with 1mM of panobinostat and belinostat unconvincing. In this case, the descipancy between MTT-based data and luminescence-based data is also rainsing up. MTT-based assay showed a dose-dependent cytotoxic effect in K562 cells treated by HDAC-inhibitors indicated above.
Another point, western blot analysis (Fig 3B) illustrates an increased expression of apoptotoc markers (cleaved forms of caspase-3 and PARP) in panbiostat-treated cells only. The authors should explain why belinostat was not effective in this case. Similarly, vorinostat had higher acetylation effect than belinostat at H3 and H4 acetylation sites (Fig 3A), but did not induce apoptosis as was shown on Fig 2.
Minor:
The quality of the western blot images shown in Fig 6 D and E should be improved (in partucular, for cleaved forms of PARP and caspase-3), no cleavage is shown there.
Just wondering, wether the different Abs for PARP were used for Fig 4B and 6E (the last oneillustrates 2 bands, the first one - the single band). No cleavage caspase (17 and 19 kDa) is shown on Fig 6Е.